# A pilot pragmatic randomized controlled trial of a 12-month Healthy Lifestyles Program: A collaborative care model for chronic conditions addressing behavioural change

Japteg Singh[1◉], David Feeny[1‡], Rebecca Ganann[2‡], John N. Lavis[1,3‡], Cynthia Lokker[4‡], Lawrence Mbuagbaw[1‡], Majdi Qutob[5‡], Zainab Samaan[6‡], Arielle Sutton[7‡], Marjan Walli-Attaei[8‡], Sarah Smith[9‡], and Elizabeth Alvarez[1◉*]

1 Department of Health Research Methods, Evidence, and Impact, McMaster University, Hamilton, Ontario, Canada, 2 School of Nursing, Faculty of Health Sciences, McMaster University, Hamilton, Ontario, Canada, 3 McMaster Health Forum, McMaster University, Hamilton, Ontario, Canada, 4 Health Information Research Unit, Department of Health Research Methods, Evidence, and Impact, McMaster University, Hamilton, Ontario, Canada, 5 Department of Surgery, McMaster University, Hamilton, Ontario, Canada, 6 Department of Psychiatry and Behavioural Neurosciences, McMaster University, Hamilton, Ontario, Canada, 7 MD Program, Schulich School of Medicine and Dentistry, Western University, London, Ontario, Canada, 8 Population Health Research Institute, McMaster University and Hamilton Health Sciences, Hamilton, Ontario, Canada, 9 St. Joseph's Healthcare Hamilton

◉ These authors contributed equally to this work.
‡ These authors also contributed equally to this work.
* alvare@mcmaster.ca

## Abstract

### Background

Lifestyle or behavioural changes can help to address the burden associated with chronic diseases. However, they take time and use of multiple techniques or strategies tailored to a person's needs. The primary objective of this study was to assess the feasibility of the Healthy Lifestyles Program (HLP), a novel 12-month complex intervention based in cognitive behavioural therapy and theories of behaviour change, delivered in a community-based setting in Hamilton, Canada. The secondary objective of the study was to explore implementation factors of the HLP.

### Methods

This pilot pragmatic randomised controlled trial used quantitative and qualitative evaluation methods. Participants were randomly allocated to either intervention group (n = 15) or comparator group (n = 15). The intervention group attended weekly group education sessions and met with the program intervention team monthly to create and review personalized health goals and action plans. The comparator group met with a trained research assistant every three months to develop health goals and action plans. We assessed program feasibility by measuring recruitment, participation

**Data availability statement:** Because of the pilot study's small sample size, data presented at the individual level would be potentially identifying for participants. For requests for de-identified and specified data access please contact the Research Institute of St. Joe's Hamilton at dcampbel@stjosham.on.ca.

**Funding:** The author(s) received no specific funding for this work.

**Competing interests:** The authors have declared that no competing interests exist.

and retention rates, missing data, and attendance. Implementation was assessed in accordance with the Reach, Effectiveness, Adoption, Implementation, Maintenance (RE-AIM) framework. Participant-directed and clinical outcome measures were analyzed for between and within group changes using Generalized Estimating Equations (GEE). Thematic analysis was conducted for qualitative data.

## Results

Retention rate was 60% (9/15) for the intervention group and 47% (7/15) for the comparator group. Less than 1% of participant-directed and clinical outcomes were missing for those that completed the study. Participants attended an average of 29 of 43 educational sessions and 100% of one-to-one sessions. The program intervention team valued the holistic approach to care, increased time and interaction with participants, professional collaboration, and the ability to provide counselling and health support. Location accessibility was an important factor facilitating implementation. Reducing the number of psycho-social education sessions and having access to a gym could improve retention and program delivery for a larger trial.

## Conclusion

This study demonstrated the feasibility of the HLP with minor modifications recommended for a larger trial and for the intervention.

## Background

Lifestyle-related chronic diseases – such as cancer, type-2 diabetes, cardiovascular and respiratory diseases – remain leading causes of preventable deaths and disability in Canada [1,2]. Many risk factors for chronic diseases are dependent on behaviour and lifestyle choices [3]. Approximately 80% of Canadian adults exhibit one or more modifiable risk factor including heavy drinking, smoking, physical inactivity, and low consumption of fruits and vegetables [1]. In Ontario, Canada, the burden associated with chronic diseases has increased over the past decade and accounts for almost 75% of all deaths and 10.5 billion dollars per year in direct healthcare costs [4]. Unfortunately, lifestyle interventions have yet to be consistently or well implemented in clinical or community-based practice settings across Ontario.

Over the past twenty years, the World Health Organization has prioritized the need to address lifestyle factors, or specific behavioural patterns that influence health outcomes, by promoting healthy lifestyle behaviours and mitigating unhealthy ones [5]. There are multiple definitions for lifestyle interventions that share a few key similarities. Broadly, lifestyle interventions include the application of evidence-based practices to help individuals adopt and sustain behavioural changes in diet, exercise, sleep, stress, substance-use, and social supports to prevent, manage and reverse lifestyle-related chronic diseases [6–8].

There are numerous barriers preventing the translation of lifestyle interventions into real-world contexts. For instance, lifestyle interventions are often designed and applied in controlled research settings that rarely evaluate process measures, which limits their applicability to pragmatic settings [9–11]. Lifestyle or behavioural interventions implemented in community settings are often short-term (less than 6 months), have reduced effect sizes, are under-funded, and often lack sufficient engagement with stakeholders, limiting their relevance (i.e., contextual and cultural) to the target population and existing networks [11,12]. Additionally, professional barriers – including lack of counselling skills, education, knowledge, and experience – limit provider's ability to sufficiently address and sustain lifestyle changes [13]. Other common barriers to implementing lifestyle interventions in routine practice include time constraints, lack of awareness of existing guidelines, lack of evidence and/or guidelines on prevention or too many guidelines for a specific purpose, communication barriers among health teams, lack of support, and patient related factors such as low adherence and negative attitudes towards prevention interventions [13].

The Healthy Lifestyles Program (HLP) is a holistic, person-centred behavioural change intervention that targets self-identified lifestyle goals to improve health outcomes through a combination of individual and group modalities. It is a program aimed at enhancing physical activity, social participation, and nutrition while addressing mental health issues that can be integrated into the individual's lifestyle and is guided by personal preferences, in contrast to a 'one-size-fits-all' approach. This program was developed from principles in Cognitive Behavioural Therapy (CBT) [14] and theories of health behaviour [15]. The HLP intervention consists of collaborative-goal setting and action-planning with program practitioners and weekly group educational sessions (both lifestyle and psychoeducational) in addition to usual care. The comparator is usual care plus participant-directed goal development and action planning with a research assistant trained in theories of health behaviour. What sets the HLP apart is its innovative integration of collaborative goal-setting, action planning, and weekly group education sessions within a single program, providing a tailored, sustainable approach to behavioural change while translating evidence-based interventions into real-world settings.

In order to evaluate the Healthy Lifestyles Program, we used the RE-AIM framework, an evaluation-implementation framework designed to translate evidence into practice, specifically in the context of scaling pragmatic studies carried out in real world, complex settings [16]. It is commonly applied to behaviour change research as it has a unique focus on process measures designed to capture intervention context, setting, and various implementation factors [16,17]. There are five domains to the Reach, Effectiveness, Adoption, Implementation and Maintenance (RE-AIM) framework that are monitored and evaluated, this includes the reach (R) of the intervention to the target population and representativeness of the target population; the effectiveness (E) of the intervention to induce change in relevant outcomes; the adoption (A) of the intervention by those responsible for implementing the intervention; the implementation (I) factors during the delivery of the intervention including consistency, cost, and adaptations and; the maintenance (M) of the intervention or sustainability of the intervention including the impact and factors related to the ability to incorporate into routine practice [18].

This study aims to determine the feasibility of a 12-month pragmatic, pilot randomized controlled trial of the novel community-based HLP for participants in Hamilton, Canada. Study findings will inform planning and scale-up of a larger, definitive trial. Therefore, the primary objectives were to assess feasibility including participant recruitment (30 participants), 12-month retention rates (≥ 50%), attendance to group and individual sessions (≥ 50%), and missing data (< 5%). The secondary objective was to assess the implementation of the HLP from the perspective of participants, program interventionists, participants' primary healthcare providers, and family members using the RE-AIM framework [19].

## Methods

### Design

This is a pragmatic two-arm, pilot randomized controlled trial that adheres to the CONSORT reporting guidelines for pilot and feasibility studies [20]. In keeping with CONSORT reporting guidelines, a CONSORT participant flow diagram is

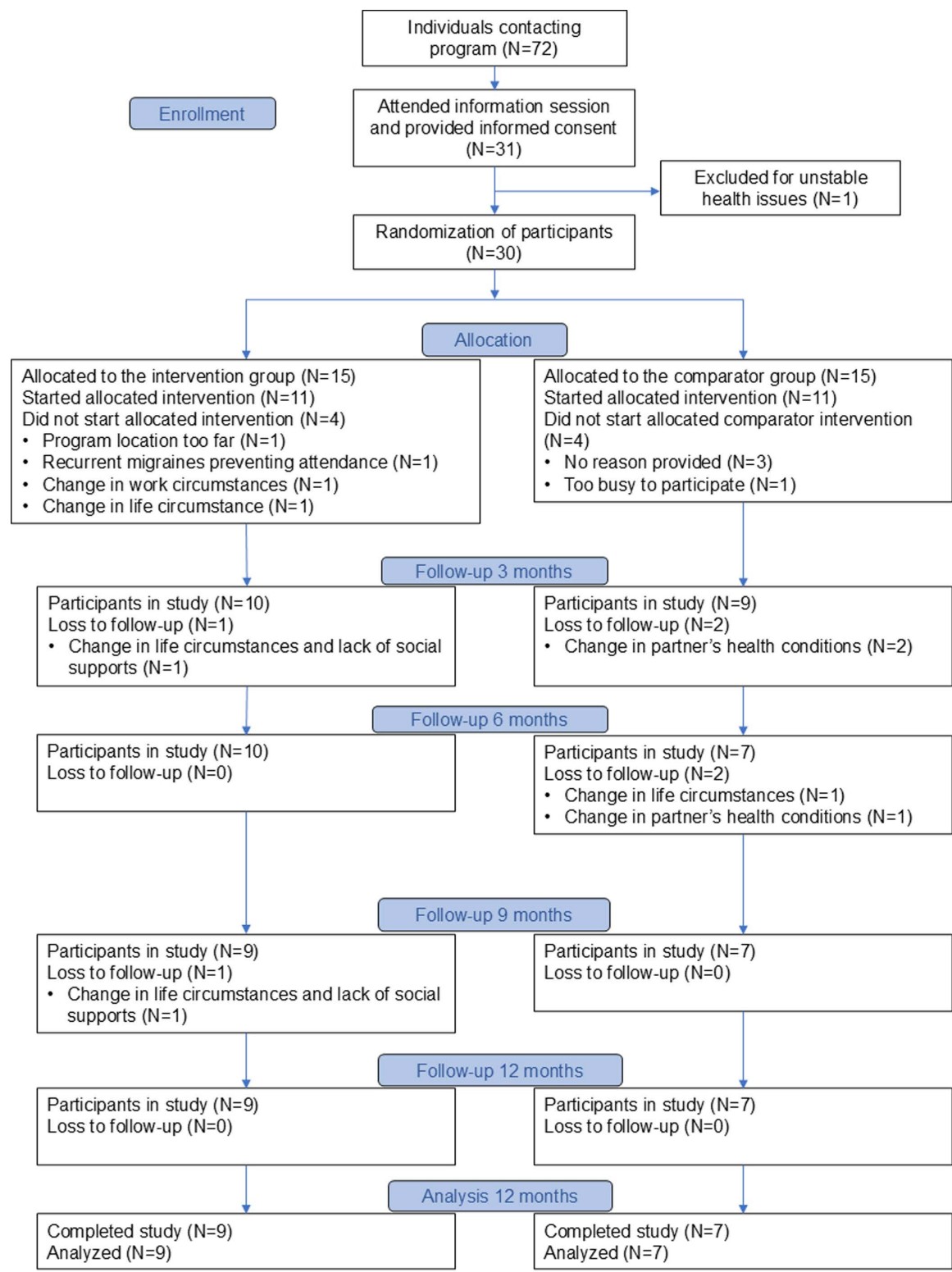

**Fig 1. Consort flow chart.**

presented for this study (Fig 1). Ethics approval was provided by the Hamilton Integrated Research Ethics Board (HiREB; #3793) and this study was registered with ClinicalTrials.gov (identifier: NCT03258138). The full details of our study protocol including design and methods are reported elsewhere [21].

## Setting, sampling, study population and recruitment

This study was conducted at McMaster University's David Braley Health Sciences Centre in Hamilton, Ontario (Canada) from April 2018 to April 2019. Participants were recruited starting 1st January 2018; enrollment was completed by 31st March 2018 and final participant outcomes were collected in April 2019.

Participants were recruited using non-probability convenience sampling through a combination of digital and print advertising. We advertised the study through Twitter, Coffee News (a free community newsletter available at local stores, restaurants, and coffee shops), and by placing recruitment posters in primary care clinics, community centres, coffee shops and office buildings across Hamilton. Interested individuals that contacted staff regarding the study were invited to an in-person information session to obtain a study overview and provide written consent. Staff followed up with participants that provided written consent to confirm eligibility prior to enrollment. The inclusion criteria were adults 18 years of age or older who were able to speak English proficiently. There were no other exclusion criteria at the outset. However, during recruitment, individuals with self-reported unstable and untreated health conditions were excluded.

Due to the pilot nature of this study, no formal sample size calculations were done. However, a sample size of 30 was chosen based on available resources, which is considered conventional for pilot studies as it permits the collection of sufficient data while minimizing research costs and allows for ideal intervention group sizes of at least 8–15 participants per group [22,23].

## Randomization

Eligible participants were assigned a 6-digit identification number and then randomly allocated to either the intervention or comparator group in a one-to-one ratio using a computer-generated randomization sequence by a research assistant who had not met the participants nor was involved in the recruitment process. The research assistant made exceptions to this randomization procedure if two or more participants indicated upon enrollment if they knew each other prior to program enrollment, in which case they were allocated to the same group based on the allocation of the participant with the lowest 6-digit identification number. This was to minimize the likelihood of cross-contamination between study groups. The 6-digit identification numbers were then uploaded to the study Research Electronic Data Capture (REDCap) database to maintain blinding during data collection and analysis.

## Healthy lifestyles program: Intervention group

The 12-month intervention was delivered using a combination of two components: in-person individualized meetings and group psycho-education sessions. The 12-month duration was selected to allow sufficient time for participants to establish and achieve health goals, identify barriers and facilitators, and implement sustainable lifestyle changes. The intervention team included a family physician trained in CBT (EA), a dietitian (SS), and an orthopaedic surgeon (MQ). Even though we had planned to have a physiotherapist at the outset, an orthopaedic surgeon was part of the intervention team because of their specialized knowledge in pain and mobility issues and due to interest in the holistic approach to care. The intervention was designed to allocate time for the interventionists to regularly meet and discuss or debrief on each participant's diagnoses, treatment, healthcare goals and action plans. Participants in the intervention group were encouraged to continue with their usual care as recommended by their physicians.

For the individualized meetings, participants met monthly in-person with the program interventionists on a one-to-one basis to develop and update their personalized healthcare goals and action plans, identify potential facilitators and barriers to their goals, and discuss strategies to modify health behaviours. For the group psycho-educational sessions, participants

were invited to attend optional weekly one-hour health and wellness learning sessions facilitated by the family physician (EA). These group sessions consisted of didactic learning, workshop presentations and open discussions on a variety of topics related to health behaviour theories and CBT to promote lifestyle changes. The sessions were designed to provide a space for social interaction whereby participants gained a level of peer-support [21]. In total, 50 weekly sessions were pre-planned with the flexibility for the session lead to adjust topics to meet participants' needs. Attendance in these sessions was not mandatory, allowing participants the flexibility to attend as many or as few sessions as they would like as would happen in real-world settings. Upon completion of the program, participants attended a group graduation ceremony and received a certificate of completion.

### Healthy lifestyles program: Comparator group

The comparator intervention was designed to mimic the types of lifestyle interventions one would expect to be delivered in real-world clinical settings as recommended by clinical practice guidelines for chronic disease (e.g., obesity, multimorbidity, etc) [24,25]. Participants allocated to the 12-month comparator group met in-person or by phone with a research assistant once every three months to develop and update their personalized healthcare goals and action plans, identify potential facilitators and barriers to their goals, and discuss various health behaviour strategies specific to their needs. The research assistant received health behaviour theories training in a graduate course at McMaster University. Participants were encouraged to continue with usual care. There were no group sessions offered to these participants though they received a certificate on completion of the program. Fig 2 illustrates the key components of the intervention and comparator groups.

### Data collection

**Feasibility: Primary objective.** S1 Table lists the descriptive and outcome measures used in this study. Feasibility was measured through recruitment, participation and retention rates, missing data, and attendance rates to individualized meetings and psycho-educational group sessions (for intervention participants). Participation rate was defined as the percent of participants who started the program after allocation, and retention rate was defined as the percent of participants who completed the program at 12-months after allocation. Participant feedback surveys were administered at three months follow-up to collect data on recruitment sources (i.e., *how did you find out about the healthy lifestyles program? list up to three)* and reasons for enrollment (i.e., *what made you want to join the program?*).

**Implementation: Secondary objective.** The implementation of the study was evaluated using the RE-AIM framework [17,26], which guided quantitative and qualitative data collection.

*Reach* was assessed by analyzing participant baseline demographic characteristics (e.g., marital status, educational level, employment status, and household income), self-reported prevalence of modifiable risk behaviours (e.g., tobacco use, recreational drug use, problematic alcohol consumption, low physical activity), overweight and obesity, and chronic diseases and multimorbidity. Problematic alcohol consumption was defined as two or more drinks per sitting for females or three or more drinks per sitting for males, low physical activity was less than 150 minutes of activity per week, overweight was a Body Mass Index (BMI) of 25–30 kg/m$^2$ and obese was defined as a BMI of 30 kg/m$^2$ or higher. Multimorbidity was defined as two or more self-reported chronic conditions. Reach outcomes were compared descriptively to Hamilton, Canada, census data [27] and Public Health Ontario's [28] health data to assess representativeness.

*Effectiveness* was assessed by analyzing participant reported outcomes for Goal Attainment Scores, anthropometric measures, mental health outcomes and participant satisfaction collected at baseline, 3, 6, 9 and 12 months. A detailed description of outcomes has been previously published in the protocol paper [21] and is listed in S1 Table. Briefly, Goal Attainment (GA) scores were measured by participants indicating their level of attainment on a 7-point Likert scale, with 1 representing the 'worst case', 7 representing the 'best case'. Each scale was personalized and developed based on the participant's ability, motivation, and purpose for the goal. Mean GA was calculated exclusively from goals created at

| Timeline | Intervention + Usual care | Comparator + Usual care |
|---|---|---|
| Randomization | | |
| Baseline | (A) (B) [D] | (C) [D] |
| 3- Month | (A) (B) [D] [E] | (C) [D] [E] |
| 6-Month | (A) (B) [D] [E] [F] [G] | (C) [D] [E] [F] [G] |
| 9- Month | (A) (B) [D] [E] [H] | (C) [D] [E] [H] |
| 12-Month | (A) (B) [D] [E] [F] [G] [I] | (C) [D] [E] [F] [G] [I] |

| | |
|---|---|
| (A) | Monthly visits with healthcare team (Primary care physician with CBT training, Dietitian, and MSK specialist) to develop goals and action plans. Community support linkages provided. |
| (B) | Weekly group psychosocial educational sessions; attendance optional. |
| (C) | 3-month visit with a research assistant to develop goals and action plans. Community support linkages provided. |
| [D] | 3-month collection of health outcomes. |
| [E] | Participant feedback survey from intervention group (N=9) and comparator group (N=7). |
| [F] | Semi-structured interview with interventionists from the intervention group (N=3) and comparator group (N=1). |
| [G] | Semi-structured interview with participant's healthcare providers (N=3). |
| [H] | Focus Group with participant family members (N=3). |
| [I] | Semi-structured interview with participants from the intervention group (N=9) and comparator group (N=7). |

◯ = program components   ☐ = Data collection/ evaluation components

**Fig 2. Overview of key components of the intervention and comparator groups.**

baseline and sustained through to the 12-month period. It was decided a priori to conduct per protocol analysis, therefore, goals developed after baseline or dropped prior to completing the 12-month study were excluded to control for the variation of time spent on goals and time enrolled in the study [29]. A change of one point was identified a priori as clinically important, as each scale measured behaviours that were participant-relevant, and each point was defined. Anthropometric measures included BMI, systolic blood pressure, diastolic blood pressure, waist circumference, hip circumference, and waist-hip ratio. Mental health outcomes were measured using validated tools; Patient Health Questionnaire-9 (PHQ-9) for depression, Generalized Anxiety Disorder 7-item scale (GAD-7) for anxiety, Insomnia Severity Index (ISI) for insomnia,

Life Change Index Scale (LCIS), and the Perceived Stress Scales 4-item (PSS4) and 10-item (PSS10) were used to measure stress. Loneliness was measured using the DeJong Gierveld Loneliness Scale 6-item questionnaire, but we modified the scoring of responses to maximize the scale's sensitivity to measure change by assigning a value of 2 to 'Yes', 1 to 'More or less' and 0 to 'No' for the positively worded items, and a value of 2 to 'No', 1 to 'More or less' and 0 to 'Yes' for the negatively worded items, changing the total scoring system from 6 to a 12-point scale [30]. The modification was made a priori to assess the degree and variability of loneliness as an issue, rather than focusing on a diagnostic categorization. Health related quality of life was measured using the RAND 36-item Short Form Health Survey (SF-36) including its composite and subscales, and the Health Utility Index Mark 2 (HUI2) and Health Utility Index Mark 3 (HUI3). Satisfaction, enjoyability and usefulness of the interventions were measured at 3, 6, and 9-months using 5-point Likert scales.

*Adoption* by those responsible to implement the intervention was assessed using qualitative data collected from semi-structured interviews with program interventionists and participant's healthcare providers at 6 and 12 months to understand why they would participate in the delivery of or refer patients to the HLP, respectively.

*Implementation* factors relating to the HLP were collected from participants, family members and program interventionists to explore their perspectives on components of the program that were implemented as planned, worked well, or could be improved. Maintenance was not assessed in this pilot study.

## Data analysis

*Quantitative Data Analysis*: All quantitative measures were analyzed using per protocol analysis to better understand the study feasibility, acceptability, and design. Demographic and feasibility outcomes were reported descriptively as counts or percentages. To analyze changes in outcome measures, generalized estimating equations (GEE) were used to estimate the average treatment effects within and between groups. The purpose of the within-group analysis was to explore change over time. Therefore, GEE analyses incorporated data from all five time-points of the study (baseline, 3, 6, 9, 12-months) separately for the intervention and comparison groups. We also conducted between group analyses at 12 months adjusting for baseline. Additionally, we used a GEE analysis to explore the correlation between attendance at group sessions in the intervention group and outcome measures. Except for the attendance and outcome analysis, all GEE models controlled for age and gender, which was decided a priori. Other covariates were not included in the analysis due to the exploratory nature of this pilot study and the small sample size. The identity link function was used for all outcomes, and the autoregressive correlation structure was used because it accounted for the longitudinal design where measurements closer in time were more closely correlated than those further apart. All quantitative data were analyzed using SPSSv27.

*Qualitative Data Analysis:* Interview transcripts were coded and analyzed in NVivo12 using inductive thematic analysis with a realist approach to interpreting the data [31,32]. Themes were coded by JS and confirmed with EA until a consensus was reached. The purpose of the qualitative data was to enrich quantitative findings, to understand implementation factors, and to enhance future iterations of the trial protocol. Qualitative feedback regarding program implementation factors were collected through semi-structured interviews with participants from the intervention (n = 9) and comparator (n = 7) groups at 12-months, program interventionists including the research assistant that delivered the comparator group (n = 4) and participant's healthcare providers (n = 3) at 6 and 12 months, focus groups with participant family members (n = 3) at 9-months, and from a participant feedback survey collected every 3-months (n = 16). All interviews were audio recorded and transcribed verbatim.

*Data Integration*: Quantitative and qualitative data were used to explore different aspects of the study. Quantitative data primarily assessed the feasibility of the study, while qualitative data provided insights into feasibility, acceptability, and the study design. Data integration was employed to understand various aspects of feasibility and implementation, with findings triangulated as appropriate to address different components. By integrating quantitative and qualitative data, this study ensured a comprehensive and holistic evaluation of the Healthy Lifestyles Program.

 

## Results

### Primary objective: Feasibility of trial protocol

*Recruitment:* Out of 31 eligible participants, 30 were recruited over three months from January to March 2018 (Fig 1). Participants were mainly recruited from local newsletter advertisements (n=8, 42%), word of mouth (n=6, 32%), and posters (n=5, 26%). Reasons for enrollment included seeking support to improve general health or lifestyle (n=12, 63%), physical health or weight loss (n=6, 32%), nutrition or healthy eating (n=2, 11%), mental health (n=1, 5%), and pain management (n=1, 5%).

*Participation and retention rates:* The participation into the study after allocation was 73% (22 of 30 participants, evenly distributed between the intervention (11/15) and comparator (11/15) groups. The overall study retention rate after 12-months was 53% (16/30); 60% (9/15) for the intervention group and 47% (7/15) for the comparator group.

*Missing data:* Less than 1% of the self-reported mental health and anthropometric data were missing for participants who completed the study after 12-months. There was no missing data for goal attainment scores. GAD-7 (anxiety) data were inconsistently collected at baseline because of technical settings in REDCap, therefore, these analyses were not included.

*Attendance to program components:* For participants who completed the study (N=16), there was 100% attendance to the individualized portions of the program in both the intervention and comparator groups. Over 50 weeks, 43 educational sessions were delivered to participants in the intervention group. Those who completed the study (n=9) attended an average of 29 (SD=7.5) optional weekly sessions with a range from 15 to 38 sessions per participant. The overall average number of participants per session was 6 (SD=1.8). Average attendance decreased over time; attendance rate was the highest in the first three months with an average of 84% and lowest in the last three months with an average of 43%.

### Secondary objective: Evaluation using RE-AIM framework

**Reach.** Table 1 summarizes baseline characteristics of participants. Four participants in the comparator group dropped out of the program after allocation and prior to collecting demographic data. Baseline data were collected at the beginning of the program to avoid overburdening the information, consent, and intake process for participants. There were no significant differences in demographic characteristics between groups. In comparison to the 2016 Hamilton census data, the program recruited a study sample of participants who were older, predominantly women, more highly educated, and less likely to work either full-time or part-time. Participants had greater prevalence of modifiable risk factors and chronic diseases, including multi-morbidity, than the general adult population of Hamilton, Ontario (Table 1).

**Effectiveness.** *Within-group analyses:* A GEE analysis was conducted to identify changes in outcomes within each group over the 12-month study period, controlling for age and gender. The results of the within-group analysis are summarized in S2 Table. Among intervention group participants, there were improvements in mean scores for insomnia, depression, stress (in both the PSS4 and PSS10 item scales), loneliness, number of active goals, goal attainment, and health-related quality of life indicators in the RAND SF-36 for pain, general health and physical composite scores over 12-months. Among comparator group participants, there were small improvements in mean scores for insomnia, perceived stress (in the PSS10 but not the 4-item scale), number of active goals, goal attainment, and RAND SF-36 general health scores over the 12-months. There were no adverse events related to the study.

*Between group analyses:* The overall test of the intervention effect compared to the comparator group was performed using a GEE analysis. S3 Table summarizes effect estimates after 12-months, adjusting for baseline differences, age, and gender. Participants in the intervention group were significantly more likely to increase their goal attainment score by 1.36 points (95%CI: 0.51, 2.20), reduce their depression score by 4.08 points (95%CI: -7.23, -0.94), reduce their loneliness scores by 2.23 points (95%CI: -3.88, -0.58), and increase their RANDSF-36 general health score by 16.92 (95%CI: 3.38, 39.47) than participants in the comparator group after 12-months. Additionally, there was a small intervention effect on decreasing waist to hip ratio after 12-months. However, these results should be interpreted as exploratory, with confidence intervals reflecting the variability and precision of estimates rather than confirming intervention effects.

**Table 1. Demographic and health characteristics of participants (N=26). Hamilton, Ontario, population data sourced from Statistics Canada and 2015-16 Hamilton data sourced from Public Health Ontario [27,28].**

| | All participants | Intervention | Comparator* | Hamilton Population |
|---|---|---|---|---|
| Total n (%) | 26 (100) | 15 (57) | 11 (43) | 536,917 |
| Mean age in years (SD) | 54.31 (12.93) | 57.87 (11.72) | 49.45 (13.45) | 41.3 |
| **Gender, n (%)** | | | | |
| Female | 21(80.7) | 12 (80) | 9 (81) | (54.5) |
| Male | 5 (19.3) | 3 (20) | 2 (19) | (45.5) |
| **Marital Status, n (%)** | | | | |
| Common Law | 3 (11.5) | 1 (7) | 2 (19) | (8.5) |
| Divorced | 4 (15.5) | 2 (13) | 2 (19) | (6.4) |
| Married | 10 (38.5) | 7 (47) | 3 (25) | (46.4) |
| Other Partner | 3 (11.5) | 1 (7) | 2 (19) | NA |
| Single | 3 (11.5) | 2 (13) | 1 (9) | (32.4) |
| Widowed | 3 (11.5) | 2 (13) | 1 (9) | (6.2) |
| **Education level, n (%)** | | | | |
| <High School | 1 (4) | 0 | 1 (9) | (17.8) |
| High School | 2 (8) | 2 (13) | 0 | (27.8) |
| College | 15 (58) | 8 (53) | 7 (63) | (22.8) |
| Bachelor's | 5 (19.5) | 4 (27) | 1 (9) | (15.5) |
| Master's | 3 (11.5) | 1 (7) | 2 (19) | (4.4) |
| **Employment status, n (%)** | | | | |
| Full-time or part-time | 12 (46) | 6 (40) | 6 (56) | (60.2) |
| Disability or sick leave | 5 (19.5) | 3 (20) | 2 (19) | NA |
| Retired | 8 (30.5) | 5 (30) | 3 (25) | NA |
| Other | 1 (2) | 1 (7) | | NA |
| **Household Income, n (%)** | | | | |
| <$20,000 | 3 (11.5) | 1 (7) | 2 (19) | (8.4) |
| $20,000–50,000 | 8 (30.5) | 5 (30) | 3 (25) | (23.6) |
| $50,001–80,000 | 5 (19.5) | 3 (20) | 2 (19) | (20.8) |
| $80,001–120, 000 | 5 (19.5) | 2 (13) | 3 (25) | (22.3) |
| >$120,000 | 4 (15.5) | 4 (27) | 0 | (24.9) |
| Did not answer | 1 (4) | 0 | 1 (9) | NA |
| | | | | % (95%CI) |
| **Modifiable Risk Behaviours, n (%)** | | | | |
| Current Tobacco Use | 2 (8) | 0 | 2 (19) | 15.3 (12.2-18.4) |
| Recreational Drug Use | 5 (19) | 1 (7) | 4 (36) | 0.09 (0.08-0.099) |
| Problematic Alcohol Consumption** | 8 (31) | 6 (40) | 2 (19) | 19.9 (16.6-23.2) |
| Low Physical Activity*** | 15 (52) | 9 (60) | 6 (56) | 22.4 (18.4-26.4) |
| Overweight+ | 7 (27) | 2 (13) | 5 (45) | 21.2 (16.9-25.6) |
| Obese+ | 17 (65) | 11(73) | 6 (56) | 34.1 (29.7-38.5) |
| **Chronic Diseases (self-reported), n (%)** | | | | |
| Anxiety | 13 (50) | 5 (33) | 8 (73) | 11 (8.3-13.7) |
| Diabetes Type 2 | 3 (11.5) | 2 (13) | 1 (9) | 7.1 (5.1-9.1) |
| Heart disease | 0 | 0 | 0 | 4.5 (3.0-6.1) |
| High blood pressure | 6 (23) | 4 (27) | 2 (19) | 20.5 (17.7–23.2) |
| Stroke | 1 (4) | 0 | 1 (9) | 1.0 (0.3-1.6) |
| Multimorbidity, n (%) | 21 (80.7) | 12 (80) | 9 (81.8) | 24.3 |

SD=Standard Deviation, NA=Not Available, CI=Confidence Interval

Percentages may not add up to 100% due to rounding.

*Missing data (N=4) in comparator group as loss to follow-up before baseline data collection

** Defined as more than 2 drinks per sitting for females or 3 drinks per sitting for males

*** Defined as below the recommended level of 150 minutes of physical activity per week

+Overweight is defined as BMI of 25 to 29.9

++Obese is defined as BMI 30 and above

*Program satisfaction:* Participants in the intervention group rated the intervention higher in satisfaction and usefulness than participants in the comparator group (Table 2). These findings were supported by the 12-month interviews with participants from both the intervention (n = 9) and comparator (n = 7) groups. Overwhelmingly, participants in the intervention group described the program as *excellent, helpful, and valuable*. Moreover, they felt the program was self-empowering, promoted health literacy or awareness, and provided the motivation and accountability to make positive changes.

**Adoption.** Adoption was assessed using qualitative interviews at 6 and 12-months from the program interventionists (n = 4) and by participant's healthcare providers (n = 3) to explore their perspectives of the program. The common themes supporting program adoption included the holistic approach to care, the ability to spend more time with participants, providing counselling and mental health supports, and professional collaboration. See Table 3 for supporting quotes.

*Holistic approach to care:* Those delivering the intervention valued the ability to address participant's symptoms and causes of complex health issues from multiple perspectives, in a collaborative care setting, as opposed to the traditional piecemeal or siloed approach they experienced in clinical settings. Providing a holistic approach to care aligned with their personal vision, mission, and goals as providers.

*Time with participants:* Program interventionists and healthcare providers valued having time to interact with their patients and identified the inability to spend time with patients as a barrier to providing care in typical clinical settings. The program interventionists valued building personal relationships with participants and being able to provide a high-quality level of care that is afforded by a 1-year program, without the external pressures and time constraints associated with a high-volume clinical practice.

*Providing counselling and mental health supports:* Program interventionists and healthcare providers felt that the lack of counselling supports around evidence-based preventative care and the promotion of healthy lifestyle behaviours were key barriers to providing behavioural change intervention in practice. Healthcare providers felt that the program's ability to provide CBT and activate patients towards healthy lifestyle behaviours were important program components that were missing in their primary care settings. The program's design to address behavioural change and provide counselling and behavioural therapy resonated with both program interventionists and healthcare providers.

*Professional communication, collaboration, and support:* The program interventionists identified the opportunity to collaborate with each other as beneficial because it afforded them the ability to provide unique input, learn from other care providers, and have an additional level of professional support that they otherwise would not have access to in their clinical practices. They felt it was meaningful to collaborate and communicate with participants to develop personalized goals and action plans, which was empowering for both the participants and the interventionists.

**Implementation.** The following were identified as key implementation factors related to program components, program length, location of program, and program resources. See Table 3 for supporting quotes.

*Program components- Individual sessions with program interventionists:* Participants highly valued the monthly in-person interaction with the program interventionists because the consistent follow-ups or check-ins allowed for timely and specialized attention to address perceived barriers in maintaining engagement towards their goals. It was important for participants to build that relationship and sense of trust through consistent interaction with the interventionist who understood their health and life context to provide the appropriate level of tailored support. Conversely, comparator group participants felt that meeting with the research assistant every three months was not enough interaction to maintain engagement towards their goals.

*Program components- Group psycho-educational sessions:* In total, 43 of the 50 planned (86%) sessions were delivered, with most sessions cancelled due to bad weather and with minor adjustments made to the programming schedule. Intervention group participants valued the group psycho-educational sessions because they felt it was an opportunity to build connections and gain support from their peers, which kept them accountable to the program. Similarly, we observed correlations between mental health and goal-related outcomes and attendance in these sessions (S4 Table). Additionally, participants found the content delivered during the sessions informative. One participant commented that it was the

**Table 2. Participant satisfaction, enjoyability and usefulness scores using a 5-point Likert scale. High values correspond to favourable outcomes for satisfaction/ enjoyability/usefulness [5 = very, 4 = somewhat, 3 = neutral, 2 = not very, 1 = not at all].**

| Scores | Intervention Group N = 9 Mean (SD) | Comparator Group N = 7 Mean | Mean Difference[a] | 95%CI |
|---|---|---|---|---|
| Satisfaction | 5.0 (0) | 3.71 (1.11) | 1.28 | 0.50 to 2.07 |
| Enjoyability | 4.89 (0.33) | 4.29 (1.11) | 0.6 | -0.23 to 1.44 |
| Usefulness | 5.0 (0) | 3.43 (1.13) | 1.57 | 0.77 to 2.37 |

[a]Independent t-test

SD- Standard Deviation; CI- Confidence Interval

combination of group sessions and individualized meetings with program interventionists that had the greatest impact on their ability to make a behavioural change. Regarding the delivery of the group psycho-educational sessions, the program interventionists commented that group sessions could be expanded to include nutritional education and cooking, yoga, and tai chi but it was also important to maintain the review or repeat sessions to re-enforce participant understanding and address behavioural change.

*Duration of program:* Participants and program interventionists felt that one year or more was an appropriate duration for the program. There was recognition that behavioural change is difficult, requires time, and seldom occurs in a linear manner. Participants felt that although health goals may be achieved within the year, it can fluctuate, and having the opportunity to work on sustaining goals was important. Seasonal fluctuations, especially winter months, were also identified as barriers to making lifestyle changes.

*Location of program:* An important implementation consideration was ensuring that the program was delivered in a space that was centrally located, accessible by public transit, had parking available, and could accommodate individuals with mobility limitations. Having a mixture of both classroom-style spaces and private consultation rooms helped facilitate the group psycho-educational sessions, as well as create a safe, private space to discuss individual health goals and action plans.

*Program resources:* Participants, family members, and interventionists suggested a few areas for program improvement. These included the need for greater program linkages to financially and physically accessible community supports and greater engagement with family members to better facilitate participants' adoption of lifestyle changes. Family members saw their role as important to providing support through encouragement, participating in behavioural changes alongside participants, and providing support outside the program. Program interventionists felt that an on-site recreational or exercise space would better address the physical health-related goals and needs for this study population to apply physical activity through practice. Program interventionists identified the need for an administrator to help facilitate program implementation including coordinating participant schedules, reminders, and logistics (e.g., room bookings, chart set-up, workflow). Having a musculoskeletal specialist was identified as an important skill set of the program interventionists as it provided for accurate diagnosis and treatment of physical and mobility related issues that would have otherwise gone untreated for this study population.

## Discussion

This pilot study evaluated the feasibility and implementation of the Healthy Lifestyles Program (HLP), a novel 12-month complex behavioural lifestyle intervention consisting of participant-centred goal development, individualized action plan setting, and group psycho-education sessions. Our findings confirm the feasibility of the HLP with minimal changes needed to conduct a larger, definitive trial. We also identified key implementation factors and documented preliminary results suggesting effectiveness for goal attainment, depression, insomnia, loneliness, and RANDSF36- general health subscale.

**Table 3. Qualitative findings using the RE-AIM framework.**

| RE-AIM Domain | Theme | Select Supporting Quote(s) |
|---|---|---|
| Effectiveness | Program Satisfaction: positive reaction of intervention program due to perceived improvements in health outcomes, attitudes and feeling of empowerment | "I think everyone should have the opportunity to have this program… because it helps you focus on your needs and wants and how to correct the habits that you accumulated all your life that are holding you back. It just is a fabulous program with your mind, your body, the exercise, it just forms a healthier you." *Intervention Participant 1–12 month* |
| | | "Last year this time I felt depressed, I was completely inactive because of pain, I was not exercising at all, I was having difficulty with stairs and getting in and out of the bathtub. I felt defeated and I felt old. And over the course of this year I felt like I was given back my life. I feel healthier, I feel stronger. I feel empowered. I feel like I have been given some very important tools. I no longer feel defeated. A big difference is in my health and my outlook. My mood, exercise and diet change have made impacts to my mental health state." *Intervention Participant 2–12 month* |
| | Program Satisfaction: mixed-reaction of comparator program due to reinforcement of health knowledge but lack of follow-up, support and accountability to sustain engagement | "I think the program re-enforced what I know, the need for exercise and the healthy benefits in doing that and the problems that I have. I still have difficulty pushing myself to do it… I needed more contact to stay on track and to help be accountable." *Comparator Participant 1–12 month* |
| | | "So, for me follow-up and support are crucial. And it is terrible to think that I can't do that for myself, but I am not good at that." *Comparator Participant 2–12 month* |
| Adoption | Holistic approach to care when addressing interconnected health concerns | "It's been extremely gratifying and enjoyable working with patients [participants] in this matter where you can actually not have to focus on one thing and be able to look at the entire person and say yes you've got knee pain that keeps you up at night that causes a lot of stress in your life." *Interventionist 3–12 month* |
| | Increased time with participants leading to more meaningful interactions between healthcare team and participants | "It's more gratifying when I don't have to try to pump them [participants] through as numbers, or not to worry about my numbers, I don't have to worry about wait times, I don't have to worry about this I can actually do something positive as opposed to say here's a prescription for this and I'll see you again in six months and then that's about it." *Interventionist 3–12 month* |
| | | "[What] struck me is how patient [centred] a program it is, that it goes on for a long time as opposed to sort of that a quick dip-in and give people some advice and set some goals and then be done. That there is follow-up and fine tuning and checking in about how things are going and that I think is realistically how people make change." *Healthcare Provider 1–6 month* |
| | Providing counselling and mental health supports | "The healthcare system is set up for acute problems and not for chronic long-term conditions and is not set up in a supportive way where you can have the counseling, or support system in place to say it's okay that you're feeling this way we can help you to a certain point, but a lot of the onus is on you as well and giving them the tools." *Interventionist 3–12 month* |
| | | "The activation piece of getting people moving and exercise and active I think as a community we don't do particularly well, and you know especially for disenfranchised people who don't have resources." *Healthcare Provider 1–6 month* |
| | | "It has been very difficult to access CBT for anybody let alone psychiatric patients. Getting into behavioural change for an active lifestyle I do not think they exist unless some of my counselors do a little bit of it on the side. So, CBT would be effective for those people [patients] but I do not know how to do it. But if I could send them to someone that would be great." *Healthcare Provider 3–12 month* |
| | Program facilitated professional communication, collaboration and support, allowing interventionists to feel valued and see their impact on participant outcomes | "I value also seeing or having other health professionals see what a [removed for identification purposes] does. You don't get that in many settings. Whereas here, when we are talking, I'm being able to showcase what my role is, how I am valued, how my suggestions are valued, and how the patients improved from these suggestions or how they are incorporating." *Interventionist 2–12 month* |
| | | "The other big thing is it is multidisciplinary…Then when we get together and talk about the patient and, all of a sudden, we see things from different angles and then we start to tease out more of the underlying issues that are involved so then we fill in each other's gaps and work really well together. Truly talking to one another about the participant, so there is a lot of communication." *Interventionist 3–6 month* |

*(Continued)*

**Table 3.** (Continued)

| RE-AIM Domain | Theme | Select Supporting Quote(s) |
|---|---|---|
| Implementation | Individual sessions with healthcare team facilitated participant accountability in the intervention group | "Well, I liked meeting once a month with the staff [interventionists] because that kept me on track and if I had barriers or obstacles and I would let them know what they are and we would work to solving them." *Intervention Participant 4–12 month* |
| | Not enough program interaction among comparator group | "Having some kind of information in between those three-month visits that was a long time and it was kind of hard to keep up the impetus I think." *Comparator participant 2–12 month* |
| | Group psycho-educational sessions facilitated opportunity to build connections and peer support | "Building the bond with people who were in the program that was also helpful to me because some days I thought oh I did not want to go because I got busy or tired I said oh no I should go because people were counting on me and/or I would learn something." *Intervention Participant 9–12 month* |
| | Group psycho-educational sessions were informative | "I feel that the most useful were the sessions because we would get handouts and we take notes and sometimes we would watch videos and staff would explain the topic in great detail so I would be able to retain the information and I had a better understanding." *Intervention Participant 4–12 month* |
| | Group psycho-educational sessions complemented individual sessions | "I think one without the other would not have been so transformational for me." *Intervention participant 3–12 month* |
| | Expanding the program to include family involvement to improve implementation | "I've been joining in activities and supportive in terms of encouraging more consistency. I do the driving so I'm supplying the transportation." *Family member 2–9 month* |
| | | "I would think the number one thing [about family] is that if they are not participating then they are not contributing. For example, I have been trying to make food changes in what I eat, but my father is not in a position where he wants to change anything that he eats. So, I think if everyone in the house is trying to do the same then there is more of a buy-in." *Comparator Participant 4–12 month* |
| | Expanding the program to include more accessible community supports | "I think there should be more help out there for people with lower income to be able to get physio and that, because I'm having difficulty getting physio that I need and being able to pay for it." *Intervention Participant 6–12 month* |
| | Physical activity or exercise was missing from the program | I think it would be great to have a location like a gym or physiotherapy where we could actually make use…so that they'll be more connection between what we're recommending and what they're ultimately doing." *Interventionist 1–12 month* |
| | | "The only thing that is lacking is exercise, actual physical doing exercise." *Intervention Participant 5–12 month* |
| | Duration of program: 12-month program provided participants with sufficient time to navigate progress and setbacks, while accounting for the effects of seasonal and personal challenges | "I think a year is really good because it is enough time that a person can experience taking steps forward and sliding back and forward again." *Interventionist 4–12 month* |
| | | "So, the one was weather or seasons, because I know I am affected with seasonal disorder. Certainly, when it is dark I do not do anything and I want to eat more or sleep more." *Comparator Participant 1–12 month* |
| | | "Yeah, I like the length of it. It was ongoing so you had your chances. I had my ups and downs and it was long winter and it was hard on everybody." *Intervention Participant 7–12 month* |
| | Location of program: centrally located and accessible | "I love the location because it was accessible, like where we get dropped off it was all level and the fact there are elevators and accessible washrooms. The fact that the room, the chairs were accessible and were not constricting." *Intervention Participant 4–12 month* |
| | Musculoskeletal (MSK) specialist and program administrator: key roles ensuring appropriate care and program delivery | "There's a huge gap in our healthcare system that does not address MSK problems. Family doctors are notoriously bad at diagnosing MSK problems and physical therapists are not always trained to diagnose … so you get this cycle where you are getting people that are being referred to orthopaedics inappropriately because it is not a surgical problem, but there should be some kind of in between non-operative orthopaedic or even sports medicine that really focuses on the MSK problems and that is one part of this program that has worked ridiculously well" *Interventionist 1–6 month* |

Passive recruitment methods [33] (e.g., local newsletter advertisements, word of mouth and posters) proved successful at enrolling the 30-participant target within three months. Although participation (73%) and retention (53%) rates after 12-months were less than the 80% threshold to limit bias in clinical trials [34], the HLP compared favourably to the retention rates observed by real-world lifestyle interventions including 6.6% for Jenny Craig program [35], 10.4% for the

Diabetes Prevention Program [35], and 26% (after 6-months) for Medifast program [36]. Achieving sufficient participation and retention rates for lifestyle trials is challenging because of the psychological, physical, and financial burdens on participants [37,38]. Seasonal factors may have contributed to the decline in attendance, as harsh winter weather during the study period may have prevented participants from attending in person. Furthermore, this program is intensive, and participants may have felt comfortable disengaging once they perceived they had gained sufficient benefits. These are salient factors for lifestyle interventions targeting mental health outcomes that require more time and effort compared to pharmacological interventions [37,38].

Higher satisfaction and usefulness scores, as well as retention rates observed in the intervention group (vs. the comparator group), indicate the intervention's acceptability and effectiveness in retaining participants with chronic mental and physical health issues. Monthly healthcare visits with a specialized team and weekly educational sessions likely contributed to this success. In contrast, the comparator group's lower retention might be due to insufficient interaction frequency of 3-months with a research assistant, leading to unmet expectations for participants motivated for lifestyle changes. This aligns with studies showing that greater program intensity often enhances efficacy [37,38].

Although not powered to detect changes in mental health or anthropometric outcomes, we did observe trends in mental health outcomes including goal attainment, depression, loneliness, and general health. We did not observe trends in improved anthropometric outcomes, except for a small change in waist to hip ratio. Lifestyle interventions alone have shown varied effects on weight management, however, when accompanied by an exercise and diet component they are likely to be more effective [39]. Additionally, because participant-directed goals were individualized and co-developed with the program interventionists, the focus of the intervention was on healthy lifestyle changes and addressing more salient risk behaviours and health outcomes like anxiety, pain management, depression, etc., to increase uptake and acceptability of the program rather than weight loss.

### Strengths and limitations

The strength of this study includes its pragmatic design and broad inclusion criteria, which resulted in a high eligibility rate of 97% and successful recruitment of participants burdened by multiple chronic diseases and modifiable risk factors that would likely seek and benefit from a lifestyle program in real-world settings. This design contrasts with many explanatory trials that have lower eligibility rates, anywhere between 16–66%, and often exclude participants with multiple chronic diseases, limiting their applicability [40,41]. This may be explained by HLP being perceived as a more holistic approach to care, which is often cited as lacking when engaging with the healthcare system that focuses on single disease management [42,43]. Other strengths include the study site's central location, accessibility to public transit and free enrollment, which facilitated the inclusion of low-income individuals. These aspects are vital for replicating the study's success in larger trials.

This study however, exhibited gender bias with 81% female participants recruited in the study, which is consistent with lifestyle or behavioural change trials that on average recruit about 27% males [44,45]. Reasons for this gender bias are not well understood but it is believed that males are less likely to exhibit health seeking behaviours towards preventative services or health promotion activities [46,47]. There is a risk of gender bias when scaling for a definitive trial, and potential mitigation strategies include using targeted recruitment methods, encouraging female participants to invite male counterparts and/or the delivery of online or self-paced group sessions to allow for some self-guided format [48,49]. Changes to the delivery of the program would need further evaluation to determine effectiveness. Furthermore, data regarding participant race and ethnicity were not collected, limiting this study's assessment of feasibility and generalizability to diverse populations. This should be incorporated in a definitive trial.

Other limitations are related to the pilot nature of the study, which include the small sample size, a single centre study, inability to blind participants, and using a comparator group that dilutes effectiveness. In the analysis, confounders such as age and gender were adjusted, however, due to the small sample size and unequal balance in gender distribution

may have introduced some spurious results. The effectiveness analysis does not control for multiplicity because the goal was to explore relationships within the data and potential impact on outcomes. Therefore, regardless of statistical significance, findings were interpreted as preliminary. All the mental health outcome measures were self-reported; however, social desirability and selective recall bias were minimized by having research assistants collect data and by using online self-administered survey tools. The blood pressure and height measurements had a great deal of variation that may limit their accuracy and precision, this was likely due to measurement error due to having multiple individuals collecting data and equipment limitations. GAD-7 scores were incomplete due to inconsistencies in the data capturing process at the beginning of the trial. Nevertheless, anxiety is an important measure that should be included in a larger trial due to the role of anxiety in managing chronic diseases.

### Implications for definitive trial

The group psycho-educational sessions were an important intervention component to build social connections and improve health outcomes, however participation did decline over time. For a larger trial, it may be worthwhile delivering these sessions using a hybrid format and/or reducing the number of sessions to match attendance trends. There was minimal missing data (<1%) for participant-centred outcomes across the 12-months, with five follow-up assessment periods. This may be attributed to the use of the REDcap system, which should be used for a larger trial to make data collection practical and to help identify incomplete records in a timely manner. Additionally, providing financial incentives during assessments may have alleviated any burdens associated with participating in the study (e.g., transportation costs) and would be beneficial for a larger trial. Furthermore, it may be beneficial to reduce the number of follow-up assessment periods to three time-points from five to reduce the resource intensive and time-consuming process of data collection.

This study was designed using a comparator group that had participants set individualized goals and action plans with the aid of a research assistant. Although the comparator group was justified, as the aim of the pilot study was to compare the intervention to what is expected in real-world clinical settings as recommended by clinical practice guidelines [24,25], it may not be feasible to continue with a comparator group for the larger trial. Furthermore, conducting a comparator or even a traditional placebo control group for an entire year is not expected to provide any further useful data, would require a considerable amount of resources and would likely underestimate the effect of the intervention [50]. This is because it is difficult to blind the subjects to control or active comparator group, likely causing participants to drop out or seek additional treatment outside of the study, and it is difficult to identify a control that is inactive but equally credible to participants [51]. For these reasons, a waitlist control group receiving usual care would be more appropriate for a larger trial.

### Conclusion

This pilot study demonstrates the feasibility of implementing the 12-month Healthy Lifestyle Program, a collaborative care person-centred behavioural change intervention. This program allows participants to self-identify goals to improve health outcomes. Trial components, individualized meetings with the intervention team, and group psycho-educational sessions, were implemented successfully with high adoption and acceptance, along with feasibility of data collection. For a larger trial it is recommended to reduce the follow-up data collection points from five to three over the 12-month period, decrease the frequency of group psycho-educational sessions to match attendance patterns or offer them in a hybrid format, and include a waitlist control group as opposed to a placebo control group.

### Supporting information

**S1 Table. Descriptive and outcome measures.**
(PDF)

**S2 Table. Within-group analysis of outcomes for participants in the intervention group (n = 9) and comparator group (n = 7) using GEE and adjusting for gender and age.**
(PDF)

**S3 Table. Between group analysis for patient-centred outcomes (N = 16) using GEE analysis and adjusting for age and gender.**
(PDF)

**S4 Table. Associations between attendance and outcomes for the intervention group participants (n = 9). Analysis using GEE and adjusting for age and gender.**
(PDF)

## Author contributions

**Conceptualization:** Lawrence Mbuagbaw, Majdi Qutob, Zainab Samaan, Sarah Smith, Elizabeth Alvarez.

**Data curation:** Japteg Singh, Arielle Sutton.

**Formal analysis:** Japteg Singh, Rebecca Ganann, Lawrence Mbuagbaw.

**Investigation:** Japteg Singh, Majdi Qutob, Sarah Smith, Elizabeth Alvarez.

**Methodology:** David Feeny, John N. Lavis, Cynthia Lokker, Lawrence Mbuagbaw, Majdi Qutob, Zainab Samaan, Arielle Sutton, Elizabeth Alvarez.

**Project administration:** Japteg Singh, Arielle Sutton, Elizabeth Alvarez.

**Supervision:** Rebecca Ganann, Lawrence Mbuagbaw, Zainab Samaan, Elizabeth Alvarez.

**Visualization:** Japteg Singh.

**Writing – original draft:** Japteg Singh.

**Writing – review & editing:** Japteg Singh, David Feeny, Rebecca Ganann, John N. Lavis, Cynthia Lokker, Lawrence Mbuagbaw, Majdi Qutob, Zainab Samaan, Arielle Sutton, Marjan Walli-Attaei, Sarah Smith, Elizabeth Alvarez.

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
