## [Decision Letter · Decision Letter 0]

26 Nov 2024

PONE-D-24-38490A Pilot Pragmatic Randomized Controlled Trial of a 12-month Healthy Lifestyles Program: A Collaborative Care Model for Chronic Conditions Addressing Behavioural ChangePLOS ONE

Dear Dr. Alvarez,

Thank you for submitting your manuscript to PLOS ONE. After careful consideration, we feel that it has merit but does not fully meet PLOS ONE’s publication criteria as it currently stands. Therefore, we invite you to submit a revised version of the manuscript that addresses the points raised during the review process.

We look forward to receiving your revised manuscript.

Kind regards,

Arkers Kwan Ching Wong, Ph.D.

Academic Editor

PLOS ONE

Journal Requirements:

3.  In this instance it seems there may be acceptable restrictions in place that prevent the public sharing of your minimal data. However, in line with our goal of ensuring long-term data availability to all interested researchers, PLOS’ Data Policy states that authors cannot be the sole named individuals responsible for ensuring data access (http://journals.plos.org/plosone/s/data-availability#loc-acceptable-data-sharing-methods).

4. Please remove your figures from within your manuscript file, leaving only the individual TIFF/EPS image files, uploaded separately. These will be automatically included in the reviewers’ PDF                .

Additional Editor Comments (if provided):

The paper requires major revision. I hope the authors can address a few things:

1. Highlight and emphasize the innovative part of this program.

2. What components were the authors intended to compare between the intervention and control groups? It seems that the authors were comparing research staff-led and healthcare professional-led interventions. This should not be the case.

3. A pilot study with the evaluation of RE-AIM may not be suitable in this study, especially the pilot study did not calculate the sample size. I would suggest the authors to try hybrid effectiveness-implementation design.

Reviewers' comments:

Reviewer's Responses to Questions

**Comments to the Author**

1. Is the manuscript technically sound, and do the data support the conclusions?

Reviewer #1: Yes

Reviewer #2: Yes

2. Has the statistical analysis been performed appropriately and rigorously? 

Reviewer #1: Yes

Reviewer #2: I Don't Know

3. Have the authors made all data underlying the findings in their manuscript fully available?

Reviewer #1: Yes

Reviewer #2: Yes

4. Is the manuscript presented in an intelligible fashion and written in standard English?

Reviewer #1: Yes

Reviewer #2: Yes

5. Review Comments to the Author

Reviewer #1: Methods

The authors indicated that during recruitment, they excluded individuals with unstable and untreated health conditions. Authors must clearly state how they identify unstable and untreated health conditions. Was it a self-reported health condition from the respondent, observation or some medical screening was done? Details must be provided

Major concern

a) The authors did not conduct any formal power analyses and decided to study a total of 30 respondents based on the rule of thumb for pilot studies. Since no power analysis was conducted, the analysis of pilot studies should be based on descriptive statistics and precision of estimates and ideally not hypothesis testing, including the use of generalized estimating equations (GEE) that report p-values and CIs for statistical inference. Pilot studies are not confirmatory trials and thus should not be hypothesis-driven. The analysis of a pilot study should be focused on descriptive statistics (e.g., means, SDs, and quantiles for continuous variables and frequencies and percentages for categorical variables) and estimates of precision. Authors indicated other covariates were not included in the analysis due to the exploratory nature of this pilot study and the small sample size but in my humble opinion, that is the more reason why the analyses should have focused on the descriptive stuff because the study was not powered to do the GEE stuff. Reporting confidence intervals of point estimates from GEE implies testing hypothesis even though p-values were not directly reported. I see not doing any formal power analyses because it is a pilot study which is ok, not controlling for other potential confounders, and only adjusting for age and sex because of smaller sample sizes, as stated by the authors, as contradictory and must be looked at. Does it also mean that the sample sample size will not have any effect on the effect size estimate if you only control for age and sex?

b) Several hypotheses were tested because the authors studied about 30 different outcomes using GEE models but did not mention how they adjusted for the effect of multiple hypothesis. If you test the effect of your Program/intervention on 30 different outcomes, and for all of them, the null hypothesis is actually true, you’d expect about one of the tests to be significant based on the confidence intervals, just due to chance because you are doing multiple hypothesis testing and NOT that the intervention actually had an effect. (more on Bonferroni correction and other more rigorous adjustment processes for multiple hypothesis testing). Not reporting p-values but reporting CI from regression model (GEE) does not mean you are not testing hypothesis.

c) There should be a separate section on the primary and secondary outcome measures and baseline characteristics, anthropometric measures, modifiable risk factors, etc, of the pilot trial. Following the entire method section is difficult and not clear because they are lumped together. In addition, the authors must clearly explain the measurement scale of these outcome measures and other variables under each sub-section (continuous, discrete, binary, multinomial, ordinal, etc.) to aid our understanding of why certain descriptive analytic measures and models were used. It may be arranged as follows:

Baseline characteristics

Primary outcome measure

Secondary outcome measures

Modifiable risk factors if it is not a primary or secondary outcome measure.

Under the sub-section, authors must clearly define the variables and how they were measured. Most of the information is already there, but merging them makes it difficult to identify key variables and the appropriateness of the model.

What was the rationale for delivering the intervention for 12-months? Authors must provide justification, Is it proven duration that the intervention can have maximum impact based on other studies, from experience or it was arbitrary? Different duration set up can lead to different impact estimate.

Authors must specifically include in the methods section the approach used to triangulate the quantitative and the qualitative findings.

Reviewer #2: Thank you for the opportunity to review this manuscript, which describes a pilot feasibility study of the Healthy Lifestyles Program (HLP), a 12-month behavioural intervention designed to address lifestyle-related chronic diseases. This study makes a meaningful contribution by addressing the critical challenge of translating evidence-based lifestyle interventions into real-world settings. The manuscript is commendable for its use of an implementation evaluation framework (RE-AIM), inclusion of an active control group, and integration of mixed methods, which enhance the study's rigor and relevance. However, greater clarity, consistency and transparency in reporting the qualitative evaluation would significantly enhance the manuscript.

Specific Areas for Improvement:

Abstract

• The abstract refers to "content analysis" (line 61) for qualitative data, while the main manuscript describes the use of "thematic analysis". This inconsistency should be resolved to ensure transparency in methodology.

Introduction

• Lines 108–109 mention that the programme was informed by CBT principles and health behaviour theories. Expanding on how these principles guided programme development would enhance reproducibility and inform the reader's understanding of the intervention, though this is not essential.

• The manuscript could benefit from a visual diagram or table to clarify the differences between the control and intervention conditions, as there is some overlap.

Methods

• The inclusion of demographic data collection is a strength of this study, highlighting the authors' attention to participant characteristics. However, ethnicity is not reported. It is unclear whether ethnicity was recorded but not analysed, or omitted entirely. Given the known disparities in health outcomes and behavioural risk factors across racial and ethnic groups, this omission limits the assessment of the intervention’s feasibility and generalisability to diverse populations. I recommend the authors clarify:

o Whether ethnicity data was collected but not reported, or not collected at all.

o The rationale for any decisions regarding ethnicity data collection or reporting.

o How future iterations of this work could address racial and ethnic disparities in both participation and outcomes to ensure equity in feasibility assessments and intervention design.

• The term "thematic analysis" appears here, contrasting with "content analysis" mentioned elsewhere. Provide a reference for the type of thematic analysis used in order to clarify the approach taken, ensuring consistency with its principles. For example, if Braun & Clarke’s thematic analysis was used, the consensus reaching approach described does not align with this methodology and divergences will need to be explained.

• The adjustment made to the scoring of the DeJong Gierveld Loneliness Scale to enhance sensitivity is an interesting and potentially valuable approach, but it may raise questions regarding its validity and comparability to other studies using the standard scoring system. Providing a brief justification for this adjustment and clarifying whether it was made a priori or post hoc would address these concerns.

Results

• A table of quotes is provided in Table 3, and the authors have performed a basic level of interpretation to connect the participant narratives to the broader RE-AIM framework. The themes identified are relevant, but some feel more descriptive than analytical (e.g., physical activity, MSK specialist). A brief explanation under each generated theme could provide greater confidence that a rigorous thematic analysis has been conducted, and help to link the participant quotes to the RE-AIM domains more explicitly.

Discussion

• It would strengthen the discussion if the authors considered potential reasons for the decrease in attendance over time. Did any of the qualitative findings indicate reasons for declining attendance or factors affecting 12-month retention rates?

• The authors attribute lower retention in the comparator group to insufficient interaction frequency (lines 485–486). While this is plausible, differences in job roles between intervention deliverers (a team of healthcare professionals) and comparator deliverers (a research assistant) should also be considered as a potential factor. Including this in the discussion would provide a more nuanced interpretation.

• If ethnicity data was indeed overlooked in this trial (see comment in methods section regarding this) then the limitations section should be expanded to include the absence of ethnicity data, which limits the evaluation of the intervention’s feasibility across diverse populations.

• In line 515, the authors propose changes to the intervention for scaling, such as online or self-paced sessions (line 515). While these ideas have merit, transitioning to self-guided formats could significantly alter the programme’s fundamental design and may not maintain the observed preliminary effectiveness. Instead, the authors should focus on exploring barriers to recruitment and retention while keeping the programme's core components intact.

Summary:

This manuscript provides valuable insights into the feasibility of implementing a collaborative care behavioural intervention for chronic conditions. The comprehensive use of the RE-AIM framework and the integration of mixed methods are particularly commendable. The study design is robust, and the results are presented clearly. However, improvements in the reporting and interpretation of the qualitative evaluation, alongside addressing some methodological and discussion points, would significantly enhance the manuscript’s overall quality.

I commend the authors for their innovative approach and for addressing an important gap in translating behavioural interventions into real-world settings. This work has great potential to inform future studies, and I hope my comments help strengthen the manuscript further.

6. PLOS authors have the option to publish the peer review history of their article (what does this mean? ). If published, this will include your full peer review and any attached files.

**Do you want your identity to be public for this peer review?** For information about this choice, including consent withdrawal, please see our Privacy Policy .

Reviewer #1: No

Reviewer #2: **Yes: ** Leanne Shearsmith

---

## [Author Response · Author response to Decision Letter 1]

31 Jan 2025

Dear Editor and reviewers,

Thank you for your thorough review of our manuscript. We respond to the comments below:

Feedback from Editor:

A data availability statement has been submitted with the documents. “Data Availability: Because of the pilot study’s small sample size, data presented at the individual level would be potentially identifying for participants. For requests for de-identified and specified data access please contact dcampbel@stjosham.on.ca.”

1. Highlight and emphasize the innovative part of this program.

Response: Thank you for this suggestion, we have emphasized the innovative aspects of this program on page 6 lines 125-128 tracked changes (page 6 lines 119-122 clean copy).

2. What components were the authors intended to compare between the intervention and control groups? It seems that the authors were comparing research staff-led and healthcare professional-led interventions. This should not be the case.

Response: Thank you for this opportunity to clarify the key components between the intervention and comparator groups. The intervention included personalized monthly visits with a healthcare team to develop health goals, identify barriers and facilitators and implement sustainable lifestyle changes with guidance from an intervention team that included a physician trained in CBT, a dietician and an orthopedic surgeon. In addition, the intervention included weekly psycho-educational sessions. In contrast, the comparator program consisted of every 3-month individual meetings to develop health goals, identify barriers and facilitators and implement sustainable lifestyle changes with the guidance of a research assistant trained in behavioural change theories. There were no psycho-educational sessional offered in the comparator program. We have included Fig 1 to illustrate the differences.

3. A pilot study with the evaluation of RE-AIM may not be suitable in this study, especially the pilot study did not calculate the sample size. I would suggest the authors to try hybrid effectiveness-implementation design.

Response: Thank you for your thoughtful feedback. While we understand that a hybrid effectiveness-implementation design may offer certain advantages, we chose the RE-AIM framework for this pilot study because of its established suitability for evaluating interventions in real-world, complex settings. The RE-AIM framework is specifically designed to assess both internal and external validity, while incorporating process measures to capture contextual, setting-specific, and implementation factors, which is critical when planning for broader scalability and real-world application.

Feedback from Reviewer 1:

1. The authors indicated that during recruitment, they excluded individuals with unstable and untreated health conditions. Authors must clearly state how they identify unstable and untreated health conditions. Was it a self-reported health condition from the respondent, observation or some medical screening was done? Details must be provided

Response: Thank you for this feedback, unstable and untreated health conditions were self reported. We clarified this on page 9 lines 174-175 tracked changes (page 9 lines 166-167 clean copy).

2. a) The authors did not conduct any formal power analyses and decided to study a total of 30 respondents based on the rule of thumb for pilot studies. Since no power analysis was conducted, the analysis of pilot studies should be based on descriptive statistics and precision of estimates and ideally not hypothesis testing, including the use of generalized estimating equations (GEE) that report p-values and CIs for statistical inference. Pilot studies are not confirmatory trials and thus should not be hypothesis-driven. The analysis of a pilot study should be focused on descriptive statistics (e.g., means, SDs, and quantiles for continuous variables and frequencies and percentages for categorical variables) and estimates of precision. Authors indicated other covariates were not included in the analysis due to the exploratory nature of this pilot study and the small sample size but in my humble opinion, that is the more reason why the analyses should have focused on the descriptive stuff because the study was not powered to do the GEE stuff. Reporting confidence intervals of point estimates from GEE implies testing hypothesis even though p-values were not directly reported. I see not doing any formal power analyses because it is a pilot study which is ok, not controlling for other potential confounders, and only adjusting for age and sex because of smaller sample sizes, as stated by the authors, as contradictory and must be looked at. Does it also mean that the sample sample size will not have any effect on the effect size estimate if you only control for age and sex?

Response: Thank you for your thoughtful feedback. We appreciate the opportunity to clarify our rationale for using Generalized Estimating Equations (GEE) in this pilot RCT. Given that this study collects repeated measures, our data are inherently correlated, making GEE a robust and appropriate method for analyzing population-averaged effects while accounting for within-subject correlation. In the context of this RCT pilot study, our primary objective was to explore feasibility and implementation of the program and the secondary objectives were to observe trends rather than conduct confirmatory hypothesis testing. The use of GEE allowed us to descriptively examine patterns, trends and variability in outcomes, providing insights to inform future, fully powered trials. We have emphasized this exploratory intent in our manuscript and have avoided overinterpreting results to ensure alignment with best practices for pilot studies. For example, on page 28 lines 593-595 tracked changes (page 28 lines 526-528 clean copy), we have stated the following in the discussion, "Although not powered to detect changes in mental health or anthropometric outcomes, we did observe trends in mental health outcomes including goal attainment, depression, loneliness, and general health ". Furthermore, we have added the following clarification sentence in the results section on page 19 lines 435-437 tracked changes (page 19 lines 391-393 clean copy)."However, these results should be interpreted as exploratory, with confidence intervals reflecting the variability and precision of estimates rather than confirming intervention effects."

2. b) Several hypotheses were tested because the authors studied about 30 different outcomes using GEE models but did not mention how they adjusted for the effect of multiple hypothesis. If you test the effect of your Program/intervention on 30 different outcomes, and for all of them, the null hypothesis is actually true, you’d expect about one of the tests to be significant based on the confidence intervals, just due to chance because you are doing multiple hypothesis testing and NOT that the intervention actually had an effect. (more on Bonferroni correction and other more rigorous adjustment processes for multiple hypothesis testing). Not reporting p-values but reporting CI from regression model (GEE) does not mean you are not testing hypothesis.

Response: Thank you for providing the opportunity to address multiplicity issues in trials. We agree that multiple testing can be a concern, particularly in confirmatory or definitive studies, where the goal is to test a specific hypothesis and where treatment effect estimates directly inform decision-making. (See: Guowei Li, Monica Taljaard, Edwin R Van den Heuvel, Mitchell AH Levine, Deborah J Cook, George A Wells, Philip J Devereaux, Lehana Thabane, An introduction to multiplicity issues in clinical trials: the what, why, when and how, International Journal of Epidemiology, Volume 46, Issue 2, April 2017, Pages 746–755, https://doi.org/10.1093/ije/dyw320)

However, in the context of exploratory trials, such as ours, the primary objective is not to provide definitive evidence but to identify potential trends to inform future, more appropriately powered confirmatory studies. Therefore, while we acknowledge the risk of type I error due to multiple hypothesis testing, adjustments for multiple comparisons are not necessary in exploratory/ pilot study designs. Instead, in this study we focus on interpreting findings with caution and transparency, recognizing that further validation is necessary. Please see notes in the above section.

3. There should be a separate section on the primary and secondary outcome measures and baseline characteristics, anthropometric measures, modifiable risk factors, etc, of the pilot trial. Following the entire method section is difficult and not clear because they are lumped together. In addition, the authors must clearly explain the measurement scale of these outcome measures and other variables under each sub-section (continuous, discrete, binary, multinomial, ordinal, etc.) to aid our understanding of why certain descriptive analytic measures and models were used. It may be arranged as follows:

Baseline characteristics

Primary outcome measure

Secondary outcome measures

Modifiable risk factors if it is not a primary or secondary outcome measure.

Under the sub-section, authors must clearly define the variables and how they were measured. Most of the information is already there, but merging them makes it difficult to identify key variables and the appropriateness of the model.

Response: Thank you for the suggestion, we have included a supplementary table, S1 Table with descriptive and outcome measures. Additionally, we have included subheadings to the data collection section to help with flow.

4. What was the rationale for delivering the intervention for 12-months? Authors must provide justification, Is it proven duration that the intervention can have maximum impact based on other studies, from experience or it was arbitrary? Different duration set up can lead to different impact estimate.

Response: Thank you, this was clarified on page 9 lines 194-196 tracked changes (page 9 lines 186-188 clean copy), “The 12-month duration was selected to allow sufficient time for participants to establish and achieve health goals, identify barriers and facilitators, and implement sustainable lifestyle changes.”

5. Authors must specifically include in the methods section the approach used to triangulate the quantitative and the qualitative findings.

Response: Thank you, we have included clarification regarding data integration in the methods on page 15 lines 334-340 tracked changes (page 15 lines 313-319 clean copy).

Feedback from Reviewer 2:

1. The abstract refers to "content analysis" (line 61) for qualitative data, while the main manuscript describes the use of "thematic analysis". This inconsistency should be resolved to ensure transparency in methodology.

Response: Thank you for bringing this to our attention, it was an oversight, and we have corrected the abstract appropriately to thematic analysis.

2. Lines 108–109 mention that the programme was informed by CBT principles and health behaviour theories. Expanding on how these principles guided programme development would enhance reproducibility and inform the reader's understanding of the intervention, though this is not essential.

Response: Thank you for highlighting the importance of expanding on how CBT principles and health behavior theories guided the program's development. As this was a pilot study, our primary focus was on assessing the feasibility and acceptability of the program, rather than creating a fully developed manual. While these principles played a foundational role in shaping the intervention, detailed refinement and documentation would be necessary before developing a comprehensive manual for broader dissemination. We added that the educational sessions included both lifestyle and psycho-educational sessions – page 6 line 123 tracked changes (page 6 line 117 clean copy).

3. The manuscript could benefit from a visual diagram or table to clarify the differences between the control and intervention conditions, as there is some overlap.

Response: Thank you, we have added Figure 1 of the key components for both the intervention and comparator groups.

4. The inclusion of demographic data collection is a strength of this study, highlighting the authors' attention to participant characteristics. However, ethnicity is not reported. It is unclear whether ethnicity was recorded but not analysed, or omitted entirely. Given the known disparities in health outcomes and behavioural risk factors across racial and ethnic groups, this omission limits the assessment of the intervention’s feasibility and generalisability to diverse populations. I recommend the authors clarify:

o Whether ethnicity data was collected but not reported, or not collected at all.

o The rationale for any decisions regarding ethnicity data collection or reporting.

o How future iterations of this work could address racial and ethnic disparities in both participation and outcomes to ensure equity in feasibility assessments and intervention design.

Response: Thank you for your thoughtful comment regarding the collection and reporting of race and ethnicity data. We acknowledge the importance of understanding health disparities across racial and ethnic groups and agree that the lack of such data in this pilot study limits the assessment of the intervention’s feasibility and generalizability to diverse populations. In this pilot study, we did not collect data on participant race or ethnicity. We recognize this as a limitation of the study, which we have added in the discussion section as a limitation. We have included the following sentences on page 29 lines 623-625 tracked changes (page 29 lines 556-558 clean copy), “Furthermore, data regarding participant race and ethnicity were not collected, limiting this study’s assessment of feasibility and generalizability to diverse populations. This should be incorporated in a definitive trial.”

5. The term "thematic analysis" appears here, contrasting with "content analysis" mentioned elsewhere. Provide a reference for the type of thematic analysis used in order to clarify the approach taken, ensuring consistency with its principles. For example, if Braun & Clarke’s thematic analysis was used, the consensus reaching approach described does not align with this methodology and divergences will need to be explained.

Response: Thank you the following citations were included to clarify the thematic analysis used:

31. Nowell, L.S., et al., Thematic Analysis: Striving to Meet the Trustworthiness Criteria. International Journal of Qualitative Methods, 2017. 16(1): p. 1609406917733847.

32. Wiltshire, G. and N. Ronkainen, A realist approach to thematic analysis: making sense of qualitative data through experiential, inferential and dispositional themes. Journal of Critical Realism, 2021. 20(2): p. 159-180.

6. The adjustment made to the scoring of the DeJong Gierveld Loneliness Scale to enhance sensitivity is an interesting and potentially valuable approach, but it may raise questions regarding its validity and comparability to other studies using the standard scoring system. Providing a brief justification for this adjustment and clarifying whether it was made a priori or post hoc would address these concerns.

Response: Thank you for the feedback. We have clarified that it was a decision made a priori and the rationale was increasing sensitivity allowed us to assess the degree and variability of loneliness rather than focusing on the diagnostic categorization of lonely vs not lonely. We have added the following clarification on page 13 lines 284-286 tracked changes (page 13 lines 270-272 clean copy), “The modification was made a priori to assess the degree and variability of loneliness as a problem, rather than focusing on a diagnostic categorization.”

7. A table of quotes is provided in Table 3, and the authors have performed a basic level of interpretation to connect the participant narratives to the broader RE-AIM framework. The themes identified are relevant, but some feel more descriptive than analytical (e.g., physical activity, MSK specialist). A brief explanation under each generated theme could provide greater confidence that a rigorous them

---

## [Decision Letter · Decision Letter 1]

17 Mar 2025

A Pilot Pragmatic Randomized Controlled Trial of a 12-month Healthy Lifestyles Program: A Collaborative Care Model for Chronic Conditions Addressing Behavioural Change

PONE-D-24-38490R1

Dear Dr. Alvarez,

We’re pleased to inform you that your manuscript has been judged scientifically suitable for publication and will be formally accepted for publication once it meets all outstanding technical requirements.

Kind regards,

Sascha Köpke

Academic Editor

PLOS ONE

Additional Editor Comments (optional):

Reviewers' comments:

Reviewer's Responses to Questions

**Comments to the Author**

1. If the authors have adequately addressed your comments raised in a previous round of review and you feel that this manuscript is now acceptable for publication, you may indicate that here to bypass the “Comments to the Author” section, enter your conflict of interest statement in the “Confidential to Editor” section, and submit your "Accept" recommendation.

Reviewer #1: All comments have been addressed

Reviewer #2: All comments have been addressed

2. Is the manuscript technically sound, and do the data support the conclusions?

Reviewer #1: Yes

Reviewer #2: Yes

3. Has the statistical analysis been performed appropriately and rigorously? 

Reviewer #1: Yes

Reviewer #2: Yes

4. Have the authors made all data underlying the findings in their manuscript fully available?

Reviewer #1: Yes

Reviewer #2: Yes

5. Is the manuscript presented in an intelligible fashion and written in standard English?

Reviewer #1: Yes

Reviewer #2: Yes

6. Review Comments to the Author

Reviewer #1: The authors have addressed all the comments from my previous review or provided clarification to concerns I raised in my earlier submission

Reviewer #2: Thank you for incorporating my feedback. I can't see a response table to reviewer comments via the system but looking at the tracked changes document I can areas that I recall highlighting have now been edited. I trust that all concerns have been addressed and therefore recommend this manuscript for publication.

7. PLOS authors have the option to publish the peer review history of their article (what does this mean? ). If published, this will include your full peer review and any attached files.

**Do you want your identity to be public for this peer review?** For information about this choice, including consent withdrawal, please see our Privacy Policy .

Reviewer #1: No

Reviewer #2: **Yes: ** Leanne Shearsmith

---

## [Editor Report · Acceptance letter]

PONE-D-24-38490R1

PLOS ONE

Dear Dr. Alvarez,

I'm pleased to inform you that your manuscript has been deemed suitable for publication in PLOS ONE. Congratulations! Your manuscript is now being handed over to our production team.

Kind regards,

on behalf of

Professor Sascha Köpke

Academic Editor

PLOS ONE